# Evaluating spatiotemporal dynamics of snakebite in Sri Lanka: Monthly incidence mapping from a national representative survey sample

**Dileepa Senajith Ediriweera** [1]*, **Anuradhani Kasthuriratne** [2]◉,
**Arunasalam Pathmeswaran** [2]◉, **Nipul Kithsiri Gunawardene** [3]◉, **Shaluka Francis Jayamanne** [4]◉, **Kris Murray** [5,6]‡, **Takuya Iwamura** [7]‡, **Geoffrey Isbister** [8,9]‡, **Andrew Dawson** [10,11]‡, **David Griffith Lalloo** [12]◉, **Hithanadura Janaka de Silva** [4]◉, **Peter John Diggle** [13]

1 Health Data Science Unit, Faculty of Medicine, University of Kelaniya, Ragama, Sri Lanka, 2 Department of Public Health, Faculty of Medicine, University of Kelaniya, Ragama, Sri Lanka, 3 Department of Parasitology, Faculty of Medicine, University of Kelaniya, Ragama, Sri Lanka, 4 Department of Medicine, Faculty of Medicine, University of Kelaniya, Ragama, Sri Lanka, 5 MRC Centre for Global Infectious Disease Analysis, Department of Infectious Disease Epidemiology, School of Public Health, Imperial College London, London, United Kingdom, 6 MRC Unit The Gambia at London School of Hygiene and Tropical Medicine, Atlantic Boulevard, Fajara, The Gambia, 7 Department of Forest Ecosystems and Society, College of Forestry, Oregon State University, Corvallis, Oregon, United States of America, 8 South Asian Clinical Toxicology Research Collaboration, University of Peradeniya, Peradeniya, Sri Lanka, 9 Clinical Toxicology Research Group, University of Newcastle, Waratah, Australia, 10 South Asian Clinical Toxicology Research Collaboration, University of Peradeniya, Peradeniya, Sri Lanka, 11 Addiction Medicine, Central Clinical School, Faculty of Medicine, University of Sydney, Sydney, Australia, 12 Liverpool School of Tropical Medicine, Liverpool, United Kingdom, 13 CHICAS, Lancaster University Medical School, Lancaster, United Kingdom

◉ These authors contributed equally to this work.
‡ KM, TI, GI and AD also contributed equally to this work.
* dileepa@kln.ac.lk

**Data Availability Statement:** All relevant data are within the paper and its Supporting Information

## Abstract

### Background

Snakebite incidence shows both spatial and temporal variation. However, no study has evaluated spatiotemporal patterns of snakebites across a country or region in detail. We used a nationally representative population sample to evaluate spatiotemporal patterns of snakebite in Sri Lanka.

### Methodology

We conducted a community-based cross-sectional survey representing all nine provinces of Sri Lanka. We interviewed 165 665 people (0.8% of the national population), and snakebite events reported by the respondents were recorded. Sri Lanka is an agricultural country; its central, southern and western parts receive rain mainly from Southwest monsoon (May to September) and northern and eastern parts receive rain mainly from Northeast monsoon (November to February). We developed spatiotemporal models using *multivariate Poisson process* modelling to explain monthly snakebite and envenoming incidences in the country.

files. The following third party data can be obtained from Census and Statistics Department of Sri Lanka (Data Dissemination Unit: data. requests@statistics.gov.lk): GN/Cluster level - population; percentage of males; percentage of agricultural workers; population mean age; percentage of people who had studied up to or above G.C.E. Advanced Level Examination; percentage of the major ethnic group. District level - mean income. The following third party data can be obtained from Department of Meteorology, Sri Lanka (Data Processing and Archival Division: metdpa@meteo.gov.lk): Monthly average rainfall, temperature and relative humidity in each bitten month. The following data can be downloaded from DIVA-GIS (http://www.diva-gis.org/gdata): Elevation.

**Funding:** This study was supported by the National Health Medical Research Council, Australia [Grant Numbers: NHMRC Project Grant 631073 obtained by GI, NHMRC Senior Research Fellowship 1061041 obtained by GI, NHMRC Practitioner Fellowship 1059542 obtained by AD and NHMRC Program Grant 1055176 obtained by GI and AD] and by the Medical Research Council, United Kingdom [MR/P024513/1 obtained by AP, KM, TI, DGL, HJdeS and PJD]. The funders had no role in study design, data collection and analysis, decision to publish, or preparation of the manuscript.

**Competing interests:** The authors have declared that no competing interests exist.

These models were developed at the provincial level to explain local spatiotemporal patterns.

## Principal findings

Snakebites and envenomings showed clear spatiotemporal patterns. Snakebite hotspots were found in North-Central, North-West, South-West and Eastern Sri Lanka. They exhibited biannual seasonal patterns except in South-Western inlands, which showed triannual seasonality. Envenoming hotspots were confined to North-Central, East and South-West parts of the country. Hotspots in North-Central regions showed triannual seasonal patterns and South-West regions had annual patterns. Hotspots remained persistent throughout the year in Eastern regions. The overall monthly snakebite and envenoming incidences in Sri Lanka were 39 (95%CI: 38–40) and 19 (95%CI: 13–30) per 100 000, respectively, translating into 110 000 (95%CI: 107 500–112 500) snakebites and 45 000 (95%CI: 32 000–73 000) envenomings in a calendar year.

## Conclusions/significance

This study provides information on community-based monthly incidence of snakebites and envenomings over the whole country. Thus, it provides useful insights into healthcare decision-making, such as, prioritizing locations to establish specialized centres for snakebite management and allocating resources based on risk assessments which take into account both location and season.

### Author summary

Snakebite envenoming is a neglected tropical disease which mainly affects people living in rural areas of the tropics. Snakebite can result in mild, moderate or severe envenomation, psychological disturbances, long-term disability, and deaths. Snakebite incidence demonstrates both spatial variation and seasonal patterns in many countries, but there have been no previous in-depth evaluations of spatiotemporal patterns of snakebite risk at a country level. We undertook an island-wide community survey to determine the spatiotemporal patterns of snakebite in the country. Both snakebites and snakebite envenomings showed clear spatiotemporal patterns where certain hotspots persisted throughout the year while others showed dynamic changes within a year. We found that the monthly snakebite incidence was 39 per 100 000 and envenoming incidences was 19 per 100 000 in Sri Lanka. We developed monthly snakebite and snakebite envenoming risk maps to identify snakebite hotspots and cold spots in the country. These maps provide useful insights into healthcare decision-making, such as, prioritizing locations to establish specialized centres for snakebite management and allocating resources based on risk assessments which consider both location and season. Our methodology demonstrates a general approach to model times and places of individual events and address the effects of variable survey effort and recall bias associated with epidemiological surveys.

## Introduction

Snakebite envenoming is a neglected tropical disease which mainly affects marginalized people living in rural areas of the tropics. Snakebites can result in mild, moderate or severe envenomation, psychological disturbances, long-term disability and deaths. Annually, 4.5–5.4 million people are bitten by snakes globally, with 1.8–2.7 million envenomings, and 81 000–138 000 deaths [1,2]. Snakebite incidence demonstrates both spatial variation and seasonal patterns in many countries [3,4], but there has been no previous in-depth evaluations of spatiotemporal patterns of snakebite risk at a country level.

Sri Lanka is an agricultural tropical island in the Indian Ocean, extending from 5˚55′ to 9˚51′N and from 79˚42′ to 81˚53′E. The geography varies from low flat plains (mean elevation 228 meters) to South-Central highland regions (mean elevation 2400 meters). Annual rainfall varies from less than 1000 mm in some areas to over 4500 mm in others, and the country is divided into three climate zones based on rainfall; wet (South-West and Central parts), dry (Northern and South-Eastern parts) and intermediate (the areas in between). Central, southern and western parts receive rain mainly from Southwest monsoon (May to September) and northern and eastern parts receive from Northeast monsoon (November to February) [5,6].

Sri Lanka has a high snakebite incidence [7]. The country has over 100 snake species. Six are medically important; *Naja naja* (cobra), *Bungarus ceylonicus* (Ceylon krait), *Bungarus caeruleus* (Common krait), *Daboia russelii* (Russell's viper), *Echis carinatus* (Saw scaled viper) and *Hypnale hypnale* (Hump nosed viper). *Naja naja* and *Daboia russelii* are widely found throughout the country. *Hypnale hypnale* can also be found throughout the country, particularly in the wet zone in rubber, tea, coconut and cocoa plantations, and are responsible for the majority of bites in the country. *Bungarus caeruleus* are found in the dry zone and *Bungarus ceylonicus* in wet mountains. Saw scaled vipers are confined to the dry arid zones of the Northern and Eastern parts of the country [8–10].

There is a clear geographical variation in the annual snakebite incidence in Sri Lanka [9]. Snakebite incidence rates have also shown seasonal patterns [10]. Therefore, snakebite is likely to demonstrate spatiotemporal variations in Sri Lanka and, to date, no study has undertaken an in-depth evaluation of spatiotemporal patterns of snakebite in the country. Identifying the high-risk geographical areas with their seasonal patterns will help to implement targeted preventive measures and allocation of resources, including antivenom, to manage the snakebite burden in Sri Lanka [11,12]. The aim of this work was to understand the spatiotemporal dynamics of snakebite in Sri Lanka in order to identify both persistent and time-varying snakebite hotspots.

## Methods

### Ethics statement

The Ethics Review Committee, Faculty of Medicine, University of Kelaniya provided the ethical approval for the study. Permission to conduct the survey in sampled clusters was obtained from the District and Divisional level public administrators and the respective *Grama Niladharis* were informed before the data collection. Written informed consent was obtained from the participants. No animals were used in the study.

### Definitions

- Snakebite is a bite event from any snake, irrespective of envenoming status.

- Envenoming bite is a snakebite with significant envenoming, which is defined as the presence of local tissue necrosis at the site of the bite, neurotoxicity or bleeding manifestations

- Bite refers to both snakebite and envenoming bite to avoid repetition in the description of the methods, as the same statistical methods were applied to model both snakebite and envenoming bites.

- Survey month is the month in which the survey was conducted. The survey was conducted for 11 consecutive survey months, August 2012 to June 2013.

- Bitten month is the month in which the victim actually experienced a snakebite. The survey reported bites occurring in 23 bitten months, August 2011 to June 2013.

- Recall time is the time difference between the bitten month and survey month of a bite. It ranged from 0 to 12 months.

### Data sources

**Epidemiological data.**   Nation-wide snakebite data were obtained from the National Snakebite Survey (NSS). This was a community-based survey designed to sample approximately 1% of the Sri Lankan population. Multistage cluster sampling was used to cover all the 9 provinces and 25 districts of the country. According to the previous national census of population and housing survey in 2001, there were estimated 18.8 million population and 3.9 million households in 18 districts with an average household size of 4.2. We estimated that there will be approximately 20 million people in 4.5 million households in Sri Lanka in 2012 (Department of Census and Statistics, Sri Lanka, 2001). Therefore, the NSS was designed to sample 45 000 households (i.e. 1% of households) in the country, consisting of 5 000 households from each of the nine provinces.

The first stage sampling units were the *Grama Niladhari* (GN) divisions which were considered as "*clusters*"for data collection. GN divisions are the smallest administrative units in the country, and Sri Lanka is divided into 14 022 GN divisions. We chose 125 clusters each from the nine provinces for the study. A province is comprised of one or more districts and the 125 clusters were proportionally allocated among the districts within the province. The number of clusters per district was selected proportional to district size. Within each district, the required number of clusters was selected by simple random sampling from the list maintained by the Department of Census and Statistics, Sri Lanka. The second stage sampling unit consisted of individual households. In each cluster, 40 households were selected for the survey, where the initial household was randomly selected based on the electoral register, which was taken as the sample frame. Proximity selection was then used to select subsequent households as the "next nearest" until the desired sample size was reached. In cases of non-response (no one in the household when the interviewers visited) the house was visited once again before the data collection in the cluster was completed. If there was still no respondent, the next nearest house in the cluster was selected.

Data were collected using an interviewer-administered two-part questionnaire. Data collection was facilitated by local volunteers. Data collectors were assisted by local field volunteers recruited within each cluster. The respondent of each household was either the head of the household or a responsible adult present in the house. Initially the research assistant screened the households for snakebite within the previous 12 months and obtained socio-demographic data from the households using the first part of the questionnaire. Subsequently, the second part of the questionnaire were administered to the households where snakebites were reported within the previous 12 months in order to obtain details of the bite (i.e. date and time, place, activity of the victim and whether the offending snake identified or not), evidence of significant envenoming, treatment details and outcome. The

questionnaire was pre-tested in each province, fine-tuned prior to use and translated into both official languages of Sri Lanka; Sinhala and Tamil. The survey was conducted for 11 consecutive months from August 2012 to June 2013. We did not consider any specific reason to select the time period for the survey [9].

**Explanatory variables.** Island-wide cluster level data on population density and percentage of agricultural workers were obtained from the Department of Census and Statistics, Sri Lanka. A rasterized elevation map for the country was obtained from DIVA-GIS [13]. The values of these variables at the centroid of each sampled cluster were considered as cluster-level spatial explanatory variables. Monthly average rainfall, temperature and relative humidity in each bitten month were obtained from the Department of Meteorology, Sri Lanka. The values of these variables at the centroid of sampled clusters in each bitten month were considered as temporal explanatory variables. The median area of a cluster is approximately 2.0 km$^2$ (IQR: 0.9–4.2) and we assumed spatial and temporal explanatory variables do not vary within a cluster.

## Statistical methods

**Spatiotemporal modelling.** We constructed separate spatiotemporal prediction models for snakebites and envenoming bites. Explanatory variables included in previously published separate spatial and temporal models for the NSS data were considered as candidate explanatory variables for the spatiotemporal models. Geostatistical binomial logistic [9] and Poisson log-linear [10] models were used to predict spatial variation in snakebite and envenoming incidences and temporal variation in national incidence, respectively.

Separate spatial and temporal models cannot explain locally varying temporal patterns in the country (i.e. persistent and time-varying snakebite hotspots). Here, we developed a spatiotemporal model for each of Sri Lanka's nine provinces, allowing for cluster level differences within each province (S1 Appendix). This was done as the survey was designed to estimate bite incidences at the provincial level and models developed at district or cluster level would have been underpowered. The NSS was conducted over 11 months and different clusters were surveyed during different months during the study period. Each cluster captured bite events that occurred over a 13-month period which included the survey month and the preceding 12 months. The NSS data contained the location of each sampled individual which was fixed at the cluster centroid, and the month of each recorded bite which varied over the 13 month period.

Our goal is to describe the rate at which bite events occur over time in each of the clusters. Our working assumption was that, in each cluster, bites occur independently of each other. As statistical model for a process of this kind is a *multivariate Poisson process*, we modelled the data from each cluster as an *inhomogenous Poisson process* with cluster-level explanatory variables. We used harmonic mathematical functions (sine and cosine terms) to model the annual, biannual, triannual and quadrennial variations in bite incidence. All the province-level models were adjusted for recall bias. This was done by including recall time as an explanatory variable in the fitted models to estimate recall bias and adjusted for this when constructing our estimates of bite incidence. Finally, the model parameters were estimated by maximising the pooled log-likelihood over each province.

To take account of the uncertainty in estimated incidence maps, we developed PCMs to identify the areas that can be confidently classified as high risk and low-risk, and areas where risk status is highly uncertain. These PCMs quantify the likelihood of exceeding or not exceeding any given threshold level of incidence at a given location and time. Probability values close to one and zero indicate geographical locations with precise classifications, whereas values

close to 0.5 indicate locations whose classification is highly uncertain. We selected the monthly mean bite incidences across all the provinces as cut off points to develop PCMs.

All computations used the R programming language version 3.2.3 [14]. Details of the statistical model fitting and prediction of monthly bite incidences are provided in the Technical appendix (S2 Appendix).

## Results

### Snakebite incidence

Different combinations of spatial variables (elevation from sea level, population density, percentage of agricultural workers in the cluster, average rainfall and temperature) appeared to be important in different provinces. Differences in harmonic terms (i.e. sine and cosine terms) showed different annual, biannual, triannual and quadrennial seasonal snakebite patterns operating in different provinces. Estimates of the recall adjustment factor corresponded to under-reporting of bites in seven of the nine provinces and two provinces where over-reporting was estimated. The two provinces in which the estimated recall factor corresponded to over-reporting had the highest and fourth highest snakebite incidences. The fitted spatiotemporal model is summarized in Table 1 (details of the parameter estimates of the fitted model is provided in S1 Table). The goodness of fit tests (S3 Appendix) and observed versus predicted snakebite incidence for the survey sample (S4 Appendix) did not show evidence of over-dispersion relative to the assumed *Poisson process model*.

**Table 1. Spatiotemporal model for snakebites.**

| Province | Spatiotemporal model for snakebite incidence |
|---|---|
| Western | - 1.65 + 0.22∗sin(12t) - 0.13∗cosine(12t) - 0.11∗sin(6t) - 0.20∗cosine(6t) + 0.01∗sin(4t) + 0.16∗cosine(4t) + 0.37∗sin(3t) - 0.21∗cosine(3t) - 0.44∗rainfall - 0.59∗temperature - 0.36∗population density + 0.31∗agriculture + 0.02∗agriculture 9% - 0.01∗recall |
| Central | - 0.41 + 0.29∗sin(12t) - 0.12∗cosine(12t) - 0.03∗sin(6t) - 0.55∗cosine(6t) + 0.02∗sin(4t) + 0.08∗cosine(4t) + 0.22∗sin(3t) - 0.33∗cosine(3t) - 1.08∗rainfall - 0.39∗population density - 0.05∗recall |
| Southern | - 9.18 + 0.06∗sin(12t) - 0.26∗cosine(12t) - 0.01∗sin(6t) - 0.30∗cosine(6t) + 0.35∗sin(4t) - 0.26∗cosine(4t) - 0.12∗sin(3t) + 0.21∗cosine(3t) - 0.37∗rainfall + 0.38∗elevation + 0.01∗recall |
| Northern | - 10.30 + 0.19∗sin(12t) + 0.07∗cosine(12t) - 0.28∗sin(6t) - 0.21∗cosine(6t) + 0.08∗sin(4t) + 0.08∗cosine(4t) + 0.03∗sin(3t) - 0.36∗cosine(3t) + 0.12∗agriculture + 0.02∗agriculture>9% - 0.06∗recall |
| Eastern | - 1.45 - 0.02∗sin(12t) - 0.33∗cosine(12t) - 0.12∗sin(6t) - 0.13∗cosine(6t) + 0.25∗sin(4t) - 0.24∗cosine(4t) - 0.28∗sin(3t) - 0.01∗cosine(3t) - 0.91∗rainfall - 0.24∗population density + 0.64∗ agriculture>9% - 0.01∗recall |
| North Western | - 3.69 + 0.01∗sin(12t) - 0.21∗cosine(12t) - 0.02∗sin(6t) - 0.14∗cosine(6t) - 0.07∗sin(4t) - 0.22∗cosine(4t) + 0.18∗sin(3t) - 0.28∗cosine(3t) + 0.67∗elevation - 1.37∗rainfall - 0.41∗agriculture - 0.02∗agriculture>9% - 0.05∗recall |
| North Central | + 0.33 + 0.02∗sin(12t) - 0.19∗cosine(12t) - 0.15∗sin(6t) - 0.21∗cosine(6t) + 0.29∗sin(4t) + 0.37∗cosine(4t) + 0.29∗sin(3t) - 0.01∗cosine(3t) - 1.37∗rainfall + 0.51∗agriculture>9% + 0.01∗recall |
| Uva | - 0.39 - 0.11∗sin(12t) - 0.01∗cosine(12t) - 0.35∗sin(6t) + 0.14∗cosine(6t) + 0.02∗sin(4t) - 0.34∗cosine(4t) - 0.06∗sin(3t) - 0.26∗cosine(3t) - 1.16∗rainfall - 0.01∗elevation>195m - 0.35∗population density - 0.01∗recall |
| Sabaragamuwa | - 3.37 - 0.20∗sin(12t) + 0.04∗cosine(12t) - 0.13∗sin(6t) - 0.05∗cosine(6t) + 0.49∗sin(4t) + 0.02∗cosine(4t) + 0.10∗sin(3t) - 0.37∗cosine(3t) - 0.70∗rainfall - 0.23∗population density - 0.04∗recall |

Sin(12t) and cosine(12t), sin(6t) and cosine(6t), sin(4t) and cosine(4t), sin(3t) and cosine(3t) denote the annual, biannual, triannual and quadrennial seasonal patterns respectively; elevation>195m denotes elevation more than 195 meters, agriculture>9% denoted the percentage of agriculture in the cluster is more than 9%.

The spatiotemporal variation in estimated monthly snakebite incidence in the whole country is shown in Fig 1. Snakebite hotspots (i.e. high-risk areas) were mainly scattered in North-Central, North-Western, South-Western and Eastern parts of the country. Most hotspots showed biannual seasonal patterns except in the South-Western inlands where triannual seasonal patterns were noticed. The hotspots were scattered over the country during January. These hotspots then developed along the North-Western and South-Western inlands during February. North-Western hotspots migrated to North-Central regions during April. South-Western hotspots geographically expanded until May. The country as a whole had a relatively low snakebite risk during June and July. In August, hotspots appeared as a band circling the central mountain regions of the country and faded away during the following two months. Thereafter, hotspots appeared in the North-Central inlands in November and then migrated to South-Western inlands during December. Hotspots in the Eastern parts appeared during September and December.

Average monthly snakebite incidence across all the provinces in Sri Lanka was 39 (95% CI: 38 – 40) bites per 100 000 people per month. This translates into 110 000 (95% CI:

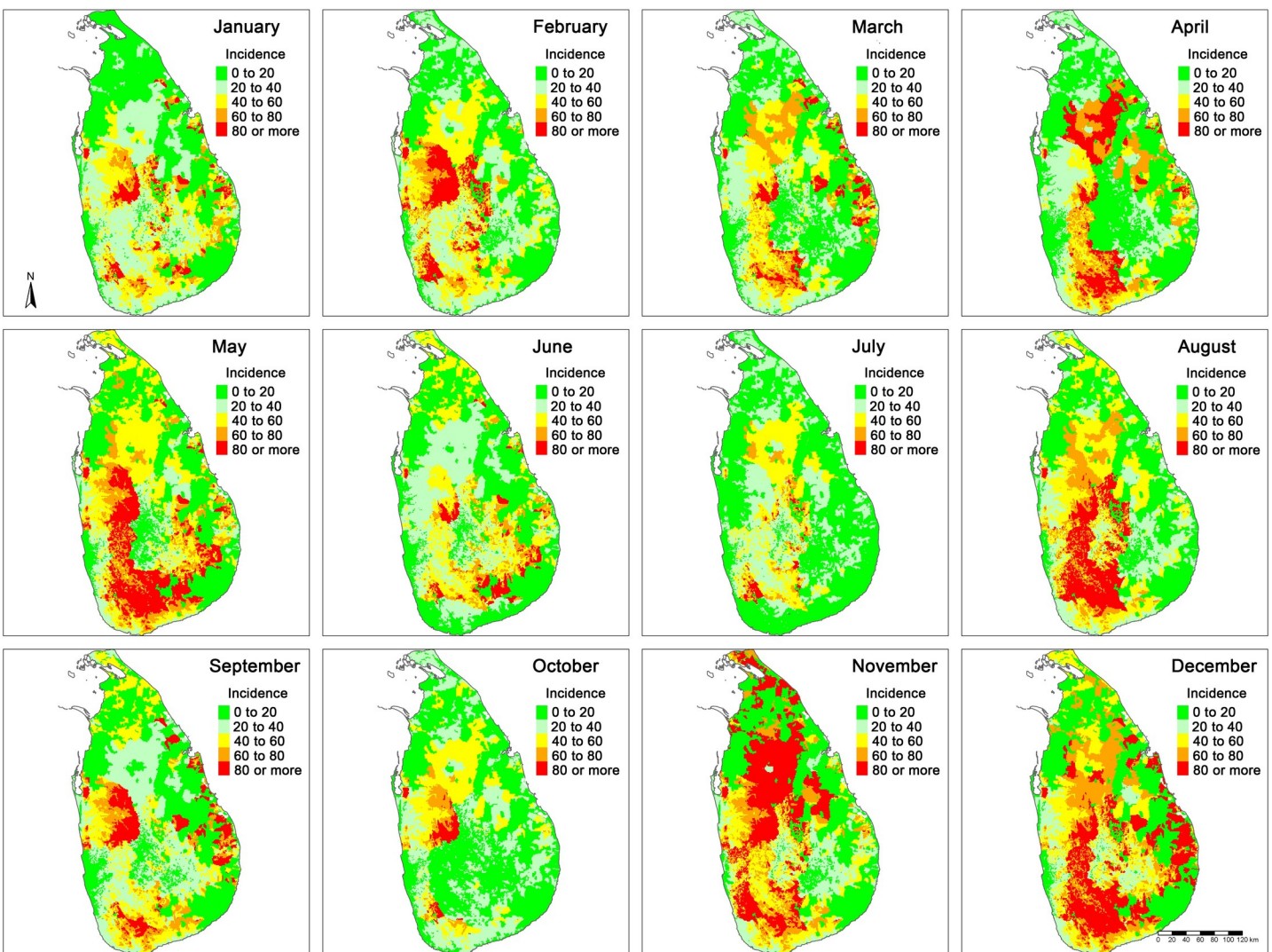

**Fig 1. Monthly snakebite incidence in Sri Lanka (bites per 100 000 people).**

107 500 – 112 500) bites per year in the whole country. The probability of exceeding 39 bites per 100 000 people in each month at any given location is shown in Fig 2. The red islands indicate the most likely places to exceed the average monthly incidence in Sri Lanka. In these areas, the probability of observing more than 39 bites per 100 000 people is at least 0.95. The blue areas have the lowest probability of exceeding the average monthly incidence. In these areas, the probability of observing more than 39 bites per 100 000 people is at most 0.05. Map areas with probability close to 0.5 indicate high uncertainty.

## Envenoming bite incidence

Different provinces showed different combinations of spatial variables (elevation from sea level, population density, percentage of agricultural workers in the cluster, average rainfall and temperature). The harmonic terms (i.e. sine and cosine terms) showed differences in provinces reflecting different annual, biannual, triannual and quadrennial seasonal snakebite patterns. Recall adjustment indicated under-reporting and over-reporting of bites in different

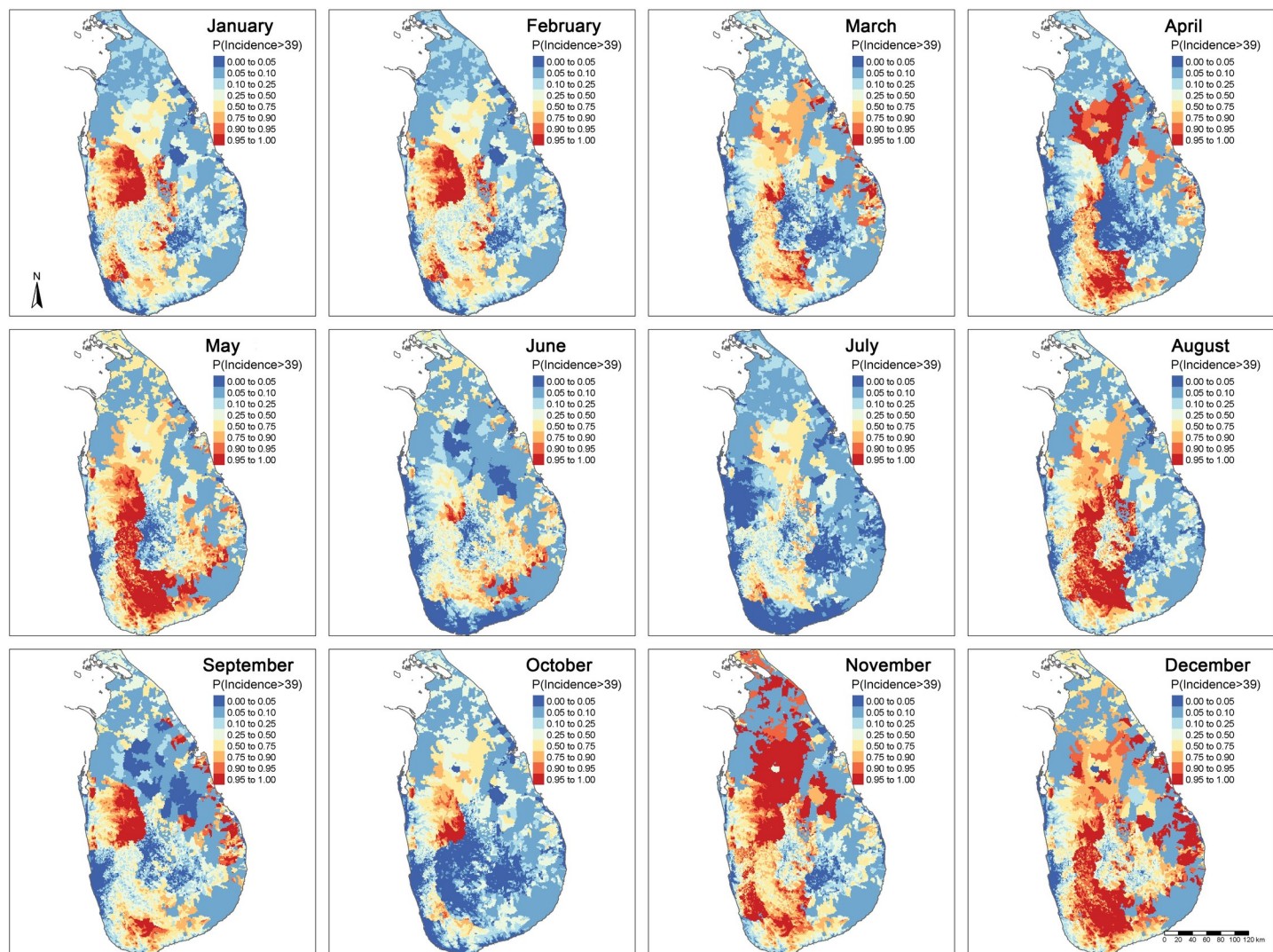

**Fig 2. Maps of exceedance probability of more than 39 snakebites per 100 000 people per month.**

provinces. The provinces with over-reporting had either high envenoming incidences or were located adjacent to provinces with high envenoming incidences. The fitted spatiotemporal model is summarized in Table 2 (details of the parameter estimates of the fitted model is provided in S2 Table). The goodness of fit tests (S5 Appendix) and observed versus predicted snakebite incidence for the survey sample (S6 Appendix) did not show evidence of over-dispersion relative to the assumed *Poisson process model*.

The spatiotemporal variation in monthly envenoming incidence in Sri Lanka is shown in Fig 3. Envenoming hotspots were mainly confined to the North-Central, Eastern and South-Western inlands of the country. Hotspots in North-Central parts showed triannual seasonal patterns that geographically expanded during March to May, July to September and November to December. Hotspots in South-Eastern regions appeared in June and then expanded towards South-Eastern regions until November. Eastern parts of the country showed high envenoming risk throughout the year.

Average monthly envenoming across all the provinces in Sri Lanka was 19 (95% CI: 13 – 30) envenomings per 100 000 people. This translates into 45 000 (95% CI: 32 000 – 73 000) envenomings in a year for the whole country. The probability of exceeding 19 bites per 100 000 people (i.e. the average monthly envenoming) in each month at any given location is shown in Fig 4. The red islands indicate the places most likely to exceed the national average monthly envenoming bite incidence. In these areas, the probability of observing more than 19 envenomings per 100 000 people is at least 0.95. The blue areas have the lowest probability to exceed the national average monthly incidence. In these areas the probability of observing more than 19 envenomings per 100 000 people is at most 0.05. Areas with probability close to 0.5 indicates areas of high uncertainty. Here, the North-Western province shows intermediate risk compared to surrounding provinces and appears as a distinct block. This is a by-product of the separate parameter estimation in each province.

**Table 2. Spatiotemporal model for envenoming bites.**

| Province | Spatiotemporal model for envenoming bite incidence |
|---|---|
| Western | - 1.36 + 0.27∗sin(12t) - 0.51∗cosine(12t) - 0.21∗sin(6t) - 0.05∗cosine(6t) + 0.27∗sin(4t) + 0.34∗cosine(4m) - 2.41∗temperature - 0.33∗population density - 0.19∗recall |
| Central | + 0.13 + 0.93∗sin(12t) - 0.34∗cosine(12t) - 0.66∗sin(6t) - 0.97∗cosine(6t) - 0.30∗sin(4t) + 0.85∗cosine(4t) -1.21∗rainfall - 0.66∗population density + 0.03∗recall |
| Southern | - 3.17 - 0.22∗sin(12t) + 0.07∗cosine(12t) - 0.28∗sin(6t) - 0.38∗cosine(6t) + 0.59∗sin(4t) - 0.29∗cosine(4t) - 2.03∗rain + 0.13∗elevation>195m + 0.04∗agriculture>9% + 0.12∗recall |
| Northern | + 0.67 + 0.07∗sin(12t) + 0.05∗cosine(12t) - 0.07∗sin(6t) - 0.23∗cosine(6t) + 0.02∗sin(4t) + 0.16∗cosine(4t) - 1.51∗rain + 0.29∗agriculture - 0.06∗recall |
| Eastern | - 4.13 + 0.21∗sin(12t) - 0.12∗cosine(12t) - 0.23∗sin(6t) + 0.11∗cosine(6t) + 0.30∗sin(4t) - 0.24∗cosine(4t) - 0.51∗rainfall - 0.30∗population density - 2.35∗agriculture + 2.64∗agriculture > 9% - 0.01∗recall |
| North Western | -5.09 + 0.26∗sin(12t) - 0.66∗cosine(12t) + 0.16∗sin(6t) - 0.09∗cosine(6t) - 0.19∗sin(4t) - 0.19∗cosine(4t) - 1.02∗rainfall + 0.16∗population density + 0.01∗recall |
| North Central | -1.92 + 0.13∗sin(12t) - 0.13∗cosine(12t) - 0.11∗sin(6t) - 0.15∗cosine(6t) + 0.19∗sin(4t) + 0.44∗cosine(4t) - 1.32∗rainfall + 0.36∗elevation + 0.08∗elevation>195m + 0.41∗agriculture>9% + 0.05∗recall |
| Uva | -0.14 - 0.12∗sin(12t) + 0.16∗cosine(12t) - 0.65∗sin(6t) + 0.14∗cosine(6t) - 0.16∗sin(4t) - 0.42∗cosine(4t) - 1.49∗rain + 0.42∗agriculture + 0.01∗recall |
| Sabaragamuwa | -5.67 - 1.36∗sin(12t) + 0.06∗cosine(12t) - 0.02∗sin(6t) + 0.56∗cosine(6t) + 0.26∗sin(4t) + 0.08∗cosine(4t) - 0.56∗rainfall + 0.72∗agriculture - 0.05∗recall |

Sin(12t) and cosine(12t), sin(6t) and cosine(6t), sin(4t) and cosine(4t) denote the annual, biannual and triannual seasonal patterns respectively; elevation>195m denotes elevation more than 195 meters, agriculture>9% denoted percentage of agriculture in the cluster is more than 9%.

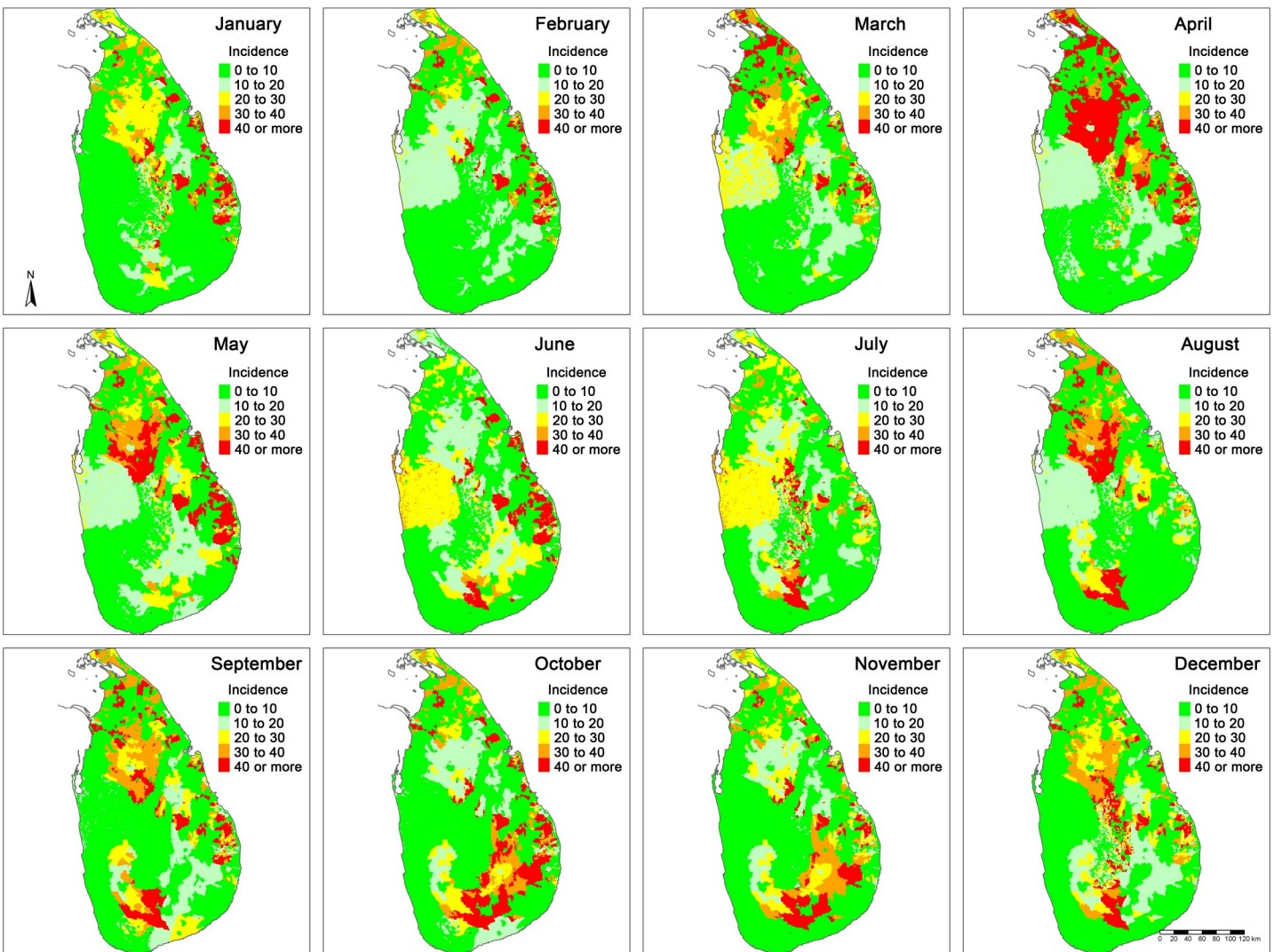

**Fig 3. Monthly envenoming bite incidence in Sri Lanka (bites per 100 000 people).**

## Discussion

Our data showed complex spatiotemporal patterns for both overall snakebites and envenomings in Sri Lanka. Snakebite hotspots were mainly observed in North-Central, North-Western, South-Western and Eastern parts of the country. These hotspots showed biannual seasonal patterns except for South-Western inlands, which had triannual patterns. Envenoming bite hotspots were mainly confined to the North-Central, Eastern and South-Western inlands of the country. Hotspots in North-Central regions showed triannual seasonal patterns. Hotspots in South-Eastern regions had annual seasonality. Hotspots in Eastern regions remained throughout the year. Western and Central provinces did not show clear spatiotemporal variation in the pattern of snakebites or envenomings. This could be due to relatively low bite incidences masking such variation. The majority of recorded bites in the South-Western region were non-envenoming. These patterns are compatible with previously reported local [15–17] and regional data [18–22]. The differences between overall snakebites and envenoming bites are likely due to the differences in the distribution of

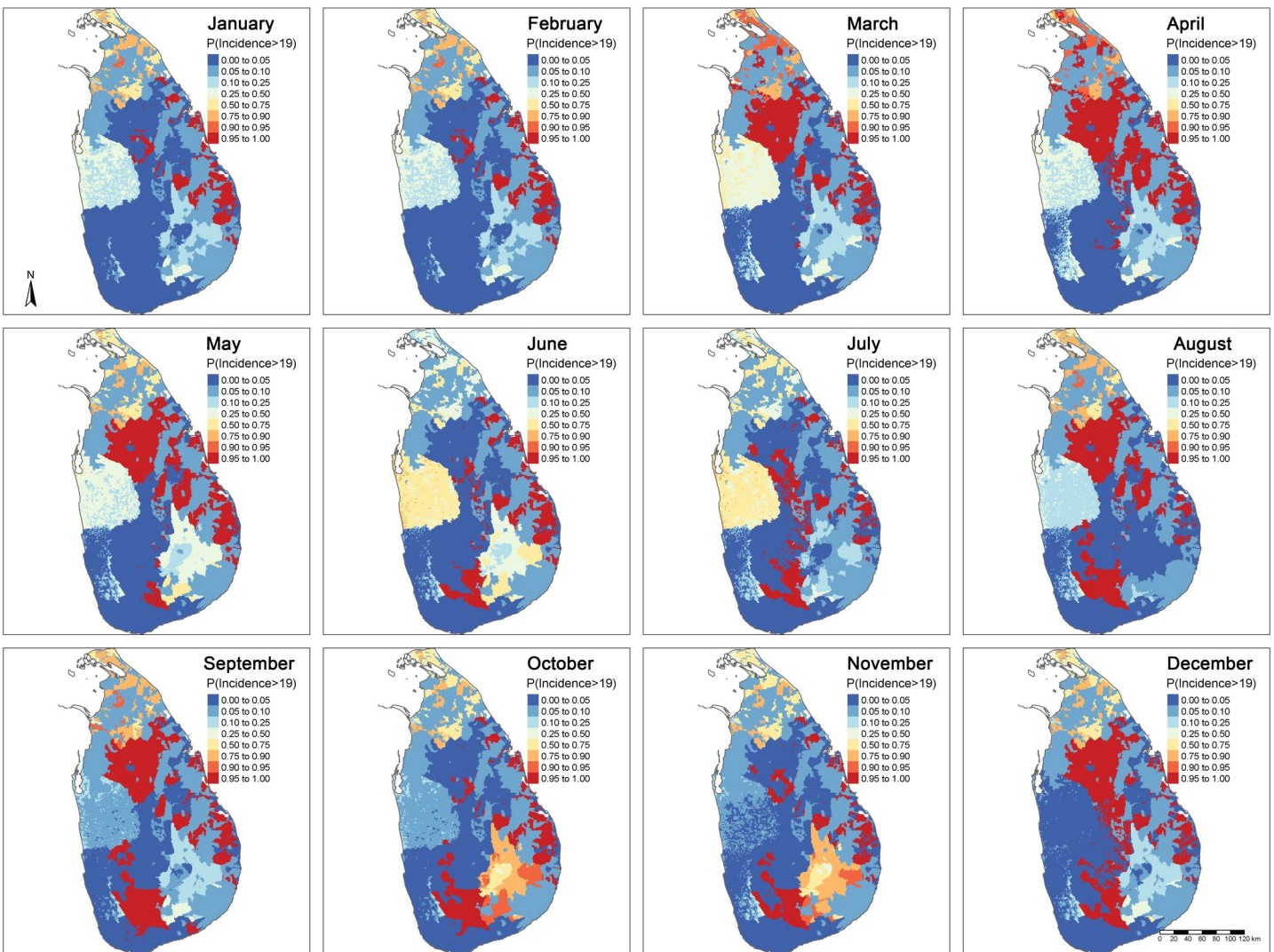

**Fig 4. Maps of exceedance probability of more than 19 envenoming bites per 100 000 people per month.**

snake species [23–28], their biological behaviour [29–33], rainfall patterns, operating agricultural seasons [6, 34] and human behaviour [35,36]. The results of our work can be used to find solutions to problems such as snakebite using a bioethical perspective. Such an approach would include discussion among the general public, academia and decision makers to identify vulnerabilities of local communities and to reach consensus on a roadmap to minimize snakebites.

We developed our spatiotemporal models at the provincial level. According to our estimates, national average snakebite and envenoming incidences were 39 and 19 per 100 000 per month, respectively. This translates into 110 000 snakebites and 45 000 envenomings in a year. The previous purely spatial analysis estimated the burden in the country to be 80 500 snakebites and 30 500 envenoming bites per year [9], whereas a purely temporal analysis estimated 119 000 snake bites per year [10]. The underestimation from a purely spatial model is likely to be due to the lack of adjustment for seasonal variations and recall bias. The overestimation from a purely temporal model is likely to be due to lack of adjustment for the spatial variation

and subnational level seasonal variations. This highlights the importance of spatiotemporal patterns when estimating disease burdens in countries.

We used harmonic mathematical functions (sine and cosine terms) to represent the seasonally varying weather (rainfall, temperature and humidity), agricultural practices and other seasonally varying phenomena in Sri Lanka [10]. In the present analysis, these harmonic functions alone were able to explain the temporal variations in snakebites and envenoming bites. None of the weather indices improved the model fits. However, other studies have indicated associations between snakebite incidence and weather indices such as rainfall [37–39]. A possible explanation for this apparent discrepancy is that harmonic functions in our models act as proxies for seasonal variation in weather-related and any other unmeasured seasonally varying correlates of snakebite incidence.

Our spatial explanatory variables (elevation from sea level, population density, percentage of agricultural workers in a cluster, long-term averaged rainfall and temperature) showed complex patterns of association with snakebite and envenoming incidences in different parts of the country. In general, incidence dropped when the population density increased, as snake habitats are less likely to coexist in population-dense areas [4]. Percentage of agricultural workers living in communities showed positive association with bite incidence, reflecting the close interaction between human and environmental factors [27]. Long-term averages of rainfall and temperature showed inverse associations with bites. Snakes are ectothermic animals and are less likely to inhabit locations subject to extreme weather conditions [29]. However, direct interpretation of the adjusted effects of variables require care. The regression coefficients in these models represent the effect of changing the values of a single variable fixing the values of all other variables. In the current context, our explanatory variables cannot typically be controlled in this way. Hence, our objective is to explain the joint effect of all the explanatory variables rather than their individual effects.

The NSS of Sri Lanka reported snakebite events experienced by family members of households in the preceding year from the date of the interview. It is possible that these reported bite counts are affected by recall bias [40]. We used an exponentially decaying function of recall duration to estimate recall bias in the fitted models and adjusted for this when constructing our estimates of snakebite incidence. The results for different provinces indicated a mix of under-reporting or over-reporting of bites. Under-reporting was estimated in provinces with low bite incidences and over-reporting in provinces with high bite incidences. A possible explanation is that in areas with high incidences, where bites are a common feature of day-to-day life, inhabitants are more likely to report a false positive memory of a bite of a previous month and vice versa [41].

Decision-making requires information of known precision. The PCMs identify the areas that can be confidently classified as high risk and low-risk, and areas where risk status is highly uncertain. Red patches on the PCMs indicate the hotspots and blue patches the cold spots, relative to the monthly mean incidences of snakebites and envenoming bites across all the provinces. These PCMs enable decision-makers to identify priority areas for establishment of specialized snakebite management centres and allocate resources, including snakebite antivenom, at operational level. Further, identification of high-risk geographical areas and seasons of snakebite are important to formulate management plans for sustainable use of natural resources. This can be achieved by environmental education processes and reduction of accidental snakebites will benefit the country including the tourism industry.

In this study, we adopted a point process modelling framework instead of the conventional regression modelling approach in order to make best use of the individual level data provided by the NSS. Times of individual events at given locations were modelled using a *multivariate Poisson process*. The NSS was conducted for 11 months and the survey sampled different

numbers of individuals at each cluster. Our methodology allowed us to estimate snakebites and envenoming bites per person incorporating spatial and temporal variables while accounting for recall bias and varying sample sizes at different clusters. The same methods could be applied for spatiotemporal modelling of other diseases for which the times and places of individual events are recorded.

There are limitations in this study. Although NSS had high spatial coverage capturing snakebite events occurring in all the provinces and districts of Sri Lanka, its temporal coverage was limited to a 13-month sampling window at each sampled location. Consequently, we were only able to disentangle seasonal trends from long-term trends by allowing for possible recall bias. Our spatiotemporal models were built from previously published separate spatial and temporal models, and the explanatory variables considered were limited to the variables used in those models. Envenoming bites are a subset of overall bites. The smaller number and fewer data-points available to fit the point process model resulted in higher standard errors and hence wider confidence intervals for envenomings than for overall snakebites. Our spatiotemporal models were constructed at the provincial level and parameter estimates were optimized for each province as the survey was designed to estimate bite incidences at provincial level. This resulted in artificial discontinuities in our fitted incidence maps at province boundaries, which are particularly apparent along the North-Western province. This can be overcome by designing future surveys at sub-province level or at district level to estimate the burden.

The present study aimed at evaluating spatiotemporal patterns of snakebite and envenoming bite in the country at the time of the survey. However, the climate change is known to affect the weather patterns of the country and hence the distribution of snakes [10]. Therefore, we intend to investigate the spatiotemporal patterns of snakebites in the changing climate, in a future study.

In conclusion, this study is the first demonstration of spatiotemporally interacting variations in snakebites and envenomings in a country. The findings can provide insights into healthcare decision-making at local level, taking into account seasonal variations, to prevent and manage snakebites. Sri Lanka could benefit by establishing a notification system for snakebites. This would provide additional documented information and augment future studies. These methods could also be applied for spatiotemporal modelling of other diseases for which the times and places of individual events are recorded.

## Supporting information

**S1 Appendix. Figure that illustrates provinces of Sri Lanka.**
(DOCX)

**S2 Appendix. Technical appendix.**
(PDF)

**S3 Appendix. Figure that illustrates reported snakebites versus predicted snakebites in each province.**
(DOCX)

**S4 Appendix. Figure that illustrates reported and predicted snakebites for the survey sample with probability integral transformation.**
(DOCX)

**S5 Appendix. Figure that illustrates reported envenoming bites versus predicted snakebites in each province.**
(DOCX)

**S6 Appendix. Figure that illustrates reported and predicted envenoming bites for the survey sample with probability integral transformation.**
(DOCX)

**S1 Table. Table that illustrates parameter estimates and standard errors in each province level for snakebites.**
(DOCX)

**S2 Table. Table that illustrated parameter estimates and standard errors in each province level for envenoming bites.**
(DOCX)

**S1 Data. Data.**
(CSV)

## Acknowledgments

The authors wish to acknowledge the assistance given by the Director General of Health Services, Provincial Directors of Health Services of the nine provinces, District Secretaries and Regional Directors of Health Services of the 25 administrative districts of Sri Lanka, all the Divisional Secretaries and the respective Grama Niladharis of the sampled Grama Niladhari Divisions, all the research managers and research assistants for implementation of the study. All the study participants are also acknowledged.

## Author Contributions

**Conceptualization:** Dileepa Senajith Ediriweera, Arunasalam Pathmeswaran, David Griffith Lalloo, Hithanadura Janaka de Silva, Peter John Diggle.

**Data curation:** Dileepa Senajith Ediriweera, Anuradhani Kasthuriratne, Nipul Kithsiri Gunawardene.

**Formal analysis:** Dileepa Senajith Ediriweera, Peter John Diggle.

**Funding acquisition:** Arunasalam Pathmeswaran, Kris Murray, Takuya Iwamura, Geoffrey Isbister, Andrew Dawson, David Griffith Lalloo, Hithanadura Janaka de Silva, Peter John Diggle.

**Investigation:** Dileepa Senajith Ediriweera, Arunasalam Pathmeswaran, David Griffith Lalloo, Hithanadura Janaka de Silva, Peter John Diggle.

**Methodology:** Dileepa Senajith Ediriweera, Peter John Diggle.

**Project administration:** Dileepa Senajith Ediriweera, Anuradhani Kasthuriratne, Nipul Kithsiri Gunawardene, Shaluka Francis Jayamanne.

**Resources:** Dileepa Senajith Ediriweera, David Griffith Lalloo, Hithanadura Janaka de Silva, Peter John Diggle.

**Software:** Dileepa Senajith Ediriweera.

**Supervision:** Arunasalam Pathmeswaran, David Griffith Lalloo, Hithanadura Janaka de Silva, Peter John Diggle.

**Validation:** Dileepa Senajith Ediriweera, Peter John Diggle.

**Visualization:** Dileepa Senajith Ediriweera, Peter John Diggle.

**Writing – original draft:** Dileepa Senajith Ediriweera, Peter John Diggle.

**Writing – review & editing:** Dileepa Senajith Ediriweera, Anuradhani Kasthuriratne, Arunasalam Pathmeswaran, Nipul Kithsiri Gunawardene, Shaluka Francis Jayamanne, Kris Murray, Takuya Iwamura, Geoffrey Isbister, Andrew Dawson, David Griffith Lalloo, Hithanadura Janaka de Silva, Peter John Diggle.

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
