## [Decision Letter · Decision Letter 0]

4 Jan 2021

Dear Dr Ediriweera,,

Thank you very much for submitting your manuscript "Evaluating spatiotemporal dynamics of snakebite in Sri Lanka: monthly incidence mapping from a national representative survey sample" for consideration at PLOS Neglected Tropical Diseases. As with all papers reviewed by the journal, your manuscript was reviewed by members of the editorial board and by several independent reviewers. In light of the reviews (below this email), we would like to invite the resubmission of a significantly-revised version that takes into account the reviewers' comments. 

We cannot make any decision about publication until we have seen the revised manuscript and your response to the reviewers' comments. Your revised manuscript is also likely to be sent to reviewers for further evaluation.

Sincerely,

Marilia Sá Carvalho

Associate Editor

José Gutiérrez

Deputy Editor

Reviewer's Responses to Questions

**Key Review Criteria Required for Acceptance?**

**Methods**

-Are the objectives of the study clearly articulated with a clear testable hypothesis stated?

-Is the study design appropriate to address the stated objectives?

-Is the population clearly described and appropriate for the hypothesis being tested?

-Is the sample size sufficient to ensure adequate power to address the hypothesis being tested?

-Were correct statistical analysis used to support conclusions?

-Are there concerns about ethical or regulatory requirements being met?

Reviewer #1: The methodology used was consistent with the results achieved.

It's all ok!

Reviewer #2: -Are the objectives of the study clearly articulated with a clear testable hypothesis stated? Need to improve.

-Is the study design appropriate to address the stated objectives? It has some critical points, but it is acceptable considering the place of study.

-Is the population clearly described and appropriate for the hypothesis being tested? I suggest that he explain better the criteria for choosing the interviewed population.

-Is the sample size sufficient to ensure adequate power to address the hypothesis being tested? Yes.

-Were correct statistical analysis used to support conclusions? Yes.

-Are there concerns about ethical or regulatory requirements being met? No.

Reviewer #3: The study shows objectives articulated with a hypothesis

The study design is appropriate

The population is clearly described

The sample size is god

There is a correct statistical analysis

The study was approved by ethical committees

**Results**

-Does the analysis presented match the analysis plan?

-Are the results clearly and completely presented?

-Are the figures (Tables, Images) of sufficient quality for clarity?

Reviewer #1: It's all ok!

Reviewer #2: -Does the analysis presented match the analysis plan? Yes.

-Are the results clearly and completely presented? It can improve a lot.

-Are the figures (Tables, Images) of sufficient quality for clarity? I found the tables a little confusing, but the images are ok.

Reviewer #3: The analysis presented does match the analysis plan; The results are clearly and completely presented and figures (Tables, Images) are relevant, with quality and clarity

**Conclusions**

-Are the conclusions supported by the data presented?

-Are the limitations of analysis clearly described?

-Do the authors discuss how these data can be helpful to advance our understanding of the topic under study?

-Is public health relevance addressed?

Reviewer #1: It's all ok!

Reviewer #2: -Are the conclusions supported by the data presented? Yes.

-Are the limitations of analysis clearly described? Yes.

-Do the authors discuss how these data can be helpful to advance our understanding of the topic under study? Yes.

-Is public health relevance addressed? Yes.

Reviewer #3: From the perspective of my role in the confluence between animals of medical and bioethical interest, I believe that there could be an innovation in the approach of the issue. I'll point out a few points. First, the authors consider it to be auxiliary in the determination of hotspots to target the service. I believe these strategies could be discussed with the community. In the matter of health control and animal accidents the community needs to see as a problem to help in the resolution. Thus, the existence of committees with popular participation, the academy and the management bodies could take advantage of the data in a more applied and real way. In this context I suggest the insertion of paradigms and perspectives of bioethics. Bioethics identifies vulnerabilities, listens to arguments and proposes solutions taken together and that are consensual and fair for all. the second point I highlight tourism, which in my perspective is an important point for the country. therefore, the knowledge of hotspots could be determinant in management plans for sustainable use of natural areas. consequently, the accidents themselves, the relationship of animal residents and the reduction of accidents could be achieved with environmental education processes. I point out that the work as it is brings an important technical contribution with data relevant to decision making, which in itself are sufficient. However, this suggestion is optional and I believe it could help bring a new perspective to the area.

**Editorial and Data Presentation Modifications?**

Reviewer #1: Insert the following reference in the correct way below:

França FGR, Braz VS.. Diversity, activity patterns, and habitat use of the snake fauna of Chapada dos Veadeiros National Park in Central Brazil. Biota Neotrop. 2013; 13:74–85. doi:10.1590/S1676-06032013000100008

Reviewer #2: I believe that the larger images would be more readable.

Reviewer #3: From the perspective of my role in the confluence between animals of medical and bioethical interest, I believe that there could be an innovation in the approach of the issue. I'll point out a few points. First, the authors consider it to be auxiliary in the determination of hotspots to target the service. I believe these strategies could be discussed with the community. In the matter of health control and animal accidents the community needs to see as a problem to help in the resolution. Thus, the existence of committees with popular participation, the academy and the management bodies could take advantage of the data in a more applied and real way. In this context I suggest the insertion of paradigms and perspectives of bioethics. Bioethics identifies vulnerabilities, listens to arguments and proposes solutions taken together and that are consensual and fair for all. the second point I highlight tourism, which in my perspective is an important point for the country. therefore, the knowledge of hotspots could be determinant in management plans for sustainable use of natural areas. consequently, the accidents themselves, the relationship of animal residents and the reduction of accidents could be achieved with environmental education processes. I point out that the work as it is brings an important technical contribution with data relevant to decision making, which in itself are sufficient. However, this suggestion is optional and I believe it could help bring a new perspective to the area.

**Summary and General Comments**

Reviewer #1: It's all ok!

Reviewer #2: Most of my considerations are in the text. I would just like to add, that if possible, make the tables easier to understand.

Reviewer #3: From the perspective of my role in the confluence between animals of medical and bioethical interest, I believe that there could be an innovation in the approach of the issue. I'll point out a few points. First, the authors consider it to be auxiliary in the determination of hotspots to target the service. I believe these strategies could be discussed with the community. In the matter of health control and animal accidents the community needs to see as a problem to help in the resolution. Thus, the existence of committees with popular participation, the academy and the management bodies could take advantage of the data in a more applied and real way. In this context I suggest the insertion of paradigms and perspectives of bioethics. Bioethics identifies vulnerabilities, listens to arguments and proposes solutions taken together and that are consensual and fair for all. the second point I highlight tourism, which in my perspective is an important point for the country. therefore, the knowledge of hotspots could be determinant in management plans for sustainable use of natural areas. consequently, the accidents themselves, the relationship of animal residents and the reduction of accidents could be achieved with environmental education processes. I point out that the work as it is brings an important technical contribution with data relevant to decision making, which in itself are sufficient. However, this suggestion is optional and I believe it could help bring a new perspective to the area.

PLOS authors have the option to publish the peer review history of their article (what does this mean?). If published, this will include your full peer review and any attached files.

Reviewer #1: Yes: Paulo Sérgio Bernarde

Reviewer #2: Yes: Gabriela Ferreira Campos Guerra

Reviewer #3: Yes: Marta Luciane Fischer
---

## [Decision Letter · Decision Letter 1]

4 May 2021

Dear Dr Ediriweera:

We are pleased to inform you that your manuscript 'Evaluating spatiotemporal dynamics of snakebite in Sri Lanka: monthly incidence mapping from a national representative survey sample' has been provisionally accepted for publication in PLOS Neglected Tropical Diseases.

Best regards,

Marilia Sá Carvalho

Associate Editor

José María Gutiérrez

Deputy Editor

<style type="text/css">p.p1 {margin: 0.0px 0.0px 0.0px 0.0px; line-height: 16.0px; font: 14.0px Arial; color: #323333; -webkit-text-stroke: #323333}span.s1 {font-kerning: none

</style>

Reviewer's Responses to Questions

**Key Review Criteria Required for Acceptance?**

**Methods**

-Are the objectives of the study clearly articulated with a clear testable hypothesis stated?

-Is the study design appropriate to address the stated objectives?

-Is the population clearly described and appropriate for the hypothesis being tested?

-Is the sample size sufficient to ensure adequate power to address the hypothesis being tested?

-Were correct statistical analysis used to support conclusions?

-Are there concerns about ethical or regulatory requirements being met?

Reviewer #2: (No Response)

Reviewer #3: The authors accepted all suggestions. The manuscript is ready to be published.

**Results**

-Does the analysis presented match the analysis plan?

-Are the results clearly and completely presented?

-Are the figures (Tables, Images) of sufficient quality for clarity?

Reviewer #2: (No Response)

Reviewer #3: The authors accepted all suggestions. The manuscript is ready to be published.

**Conclusions**

-Are the conclusions supported by the data presented?

-Are the limitations of analysis clearly described?

-Do the authors discuss how these data can be helpful to advance our understanding of the topic under study?

-Is public health relevance addressed?

Reviewer #2: (No Response)

Reviewer #3: The authors accepted all suggestions. The manuscript is ready to be published.

**Editorial and Data Presentation Modifications?**

Reviewer #2: (No Response)

Reviewer #3: The authors accepted all suggestions. The manuscript is ready to be published.

**Summary and General Comments**

Reviewer #2: (No Response)

Reviewer #3: The authors accepted all suggestions. The manuscript is ready to be published.

PLOS authors have the option to publish the peer review history of their article (what does this mean?). If published, this will include your full peer review and any attached files.

Reviewer #2: **Yes: **Gabriela Ferreira Campos Guerra

Reviewer #3: **Yes: **...

---

## [Editor Report · Acceptance letter]

26 May 2021

Dear Dr Ediriweera,

We are delighted to inform you that your manuscript, "Evaluating spatiotemporal dynamics of snakebite in Sri Lanka: monthly incidence mapping from a national representative survey sample," has been formally accepted for publication in PLOS Neglected Tropical Diseases.

Best regards,

Shaden Kamhawi

co-Editor-in-Chief

Paul Brindley

co-Editor-in-Chief
